# The Impact of SARS-CoV-2 Primary Vaccination in a Cohort of Patients Hospitalized for Acute COVID-19 during Delta Variant Predominance

**DOI:** 10.3390/jcm11051191

**Published:** 2022-02-23

**Authors:** Daša Stupica, Stefan Collinet-Adler, Nataša Kejžar, Zala Jagodic, Mario Poljak, Mirijam Nahtigal Klevišar

**Affiliations:** 1Department of Infectious Diseases, University Medical Center Ljubljana, Japljeva 2, 1525 Ljubljana, Slovenia; mirijam.nahtigal@kclj.si; 2Department of Infectious Diseases, Faculty of Medicine, University of Ljubljana, Vrazov trg 2, 1000 Ljubljana, Slovenia; zala.jagodic@gmail.com; 3Department of Infectious Diseases, Methodist Hospital, Park Nicollet/Health Partners, Minneapolis, MN 55426, USA; stefan.collinet_adler@parknicollet.com; 4Institute for Biostatistics and Medical Informatics, Faculty of Medicine, University of Ljubljana, Vrazov trg 2, 1000 Ljubljana, Slovenia; natasa.kejzar@mf.uni-lj.si; 5Institute of Microbiology and Immunology, Faculty of Medicine Ljubljana, Zaloška 4, 1000 Ljubljana, Slovenia; mario.poljak@mf.uni-lj.si

**Keywords:** COVID-19 vaccine, vaccine breakthrough, COVID-19 outcome

## Abstract

Vaccine breakthrough SARS-CoV-2 infections necessitating hospitalization have emerged as a relevant problem with longer time interval since vaccination and the predominance of the Delta variant. The aim of this study was to evaluate the association between primary vaccination with four SARS-CoV-2 vaccines authorized for use in the European Union—BNT162b2, ChAdOx-1S, mRNA-1273 or Ad.26.COV2.S—and progression to critically severe disease (mechanical ventilation or death) and duration of hospitalization among adult patients with PCR-confirmed acute COVID-19 hospitalized during the Delta variant predominance (October–November 2021) in Slovenia. Among the 529 enrolled patients hospitalized with COVID-19 (median age, 65 years; 58.2% men), 175 (33.1%) were fully vaccinated at the time of symptom onset. Compared with 345 unvaccinated patients, fully vaccinated patients with breakthrough infections were older, more often immunocompromised, and had higher Charlson comorbidity index scores. After adjusting for sex, age, and comorbidities, fully vaccinated patients had lower odds for progressing to critically severe disease and were discharged from the hospital earlier than unvaccinated patients. Vaccination against SARS-CoV-2 remains an extremely effective intervention to alleviate morbidity and mortality in COVID-19 patients.

## 1. Introduction

Prior infection with severe acute respiratory syndrome coronavirus 2 (SARS-CoV-2) generates humoral and cellular correlates of immunity [1,2,3,4] and some protection of uncertain duration from re-infection [5,6,7]. However, the permissive accumulation of infections in populations is a poor strategy for controlling the pandemic given the considerable morbidity and mortality of coronavirus disease 2019 (COVID-19). To face this public health challenge, many different types of vaccines against SARS-CoV-2 have been explored using various vaccine platforms, including non-replicating adenoviral vector-based and mRNA strategies [8]. The development, licensing, and distribution of several highly safe and effective vaccines against SARS-CoV-2 have transformed the trajectory of the pandemic by providing significant protection from COVID-19 [9,10,11,12]. Nevertheless, breakthrough SARS-CoV-2 infections necessitating hospitalization have emerged as a relevant problem with predominance of the Delta variant followed by the Omicron variant and longer time interval since vaccination [13,14,15]. Among the patients hospitalized with COVID-19 between 11 March and 15 August 2021 in the US, a period during which the Alpha variant was progressively replaced by the Delta variant, vaccination with mRNA COVID-19 vaccines was still associated with reduced risk of progression to invasive mechanical ventilation or death compared with the absence of vaccination [14]. Given the complex interplay of pandemic dynamics with the Delta variant predominating after the Alpha variant coupled with waning immunity over time, it is not entirely clear if or to what extent differential efficacy of vaccines against different strains exist [14]. Protection against emergency department or urgent care visits and hospitalization differ between the various SARS-CoV-2 vaccines [16]. In particular, BNT162b2 (Pfizer-BioNTech) was associated with less protection from illness requiring hospitalization than mRNA-1273 (Moderna) beyond 120 days after vaccination [14].

The aim of this study was to evaluate the association between primary vaccination with four SARS-CoV-2 vaccines authorized for use in the European Union—BNT162b2, ChAdOx-1S (AstraZeneca), mRNA-1273 or Ad.26.COV2.S (Janssen)—and progression to critically severe disease as well as duration of hospitalization among the patients hospitalized with COVID-19 during a period with Delta variant predominance.

## 2. Materials and Methods

### 2.1. Design and Setting

A total of 566 adult patients (≥18 years) hospitalized with COVID-19 at a single university medical center in Ljubljana, Slovenia between 1 October and 23 November 2021, were assessed for enrollment on 23 December 2021. STROBE guidelines for reporting were followed. We assessed the association between COVID-19 disease severity and length of hospitalization with prior COVID-19 vaccination.

### 2.2. Participants

Hospitalized patients were screened for potential eligibility through the hospital electronic medical record system. Patients were deemed eligible if they were hospitalized with a clinical presentation consistent with acute COVID-19 and had a positive SARS-CoV-2 PCR test within 10 days before admission [17]. Patients who were hospitalized for medical conditions unrelated to COVID-19 and patients partially vaccinated against SARS-CoV-2 were excluded.

### 2.3. Data Collection

Demographic, clinical, and laboratory data were collected through the hospital medical record review. Dates of SARS-CoV-2 vaccination, vaccine product, and lot numbers were verified using the national electronic vaccine registry.

### 2.4. Laboratory Analysis

The presence of SARS-CoV-2 RNA in upper respiratory specimens were determined using one of two fully automated rtRT-PCR analyzers: Cobas 6800 (Roche Diagnostics, Basel, Switzerland) or Alinity m (Abbott, Chicago, IL, USA), as described previously [18].

The presence of total antibodies to the SARS-CoV-2 spike (S) protein receptor binding domain was determined using automated electrochemiluminescence immunoassay Elecsys Anti-SARS-CoV-2 S (Roche Diagnostics), as described previously [19]. Results were interpreted according to the manufacturer’s instructions.

### 2.5. Classification of Vaccination Status and COVID-19 Severity

Patients were considered fully vaccinated if they received the second dose of BNT162b2, mRNA-1273 or ChAdOx-1S or first dose of Ad.26.COV2.S at least 14 days before symptom onset [20]. Participants were classified as unvaccinated if they had received no vaccine doses before symptom onset or partially vaccinated if they had received one dose of BNT162b2 or mRNA-1273 or two doses of BNT162b2 or mRNA-1273 or one dose of Ad.26.COV2.S within 14 days before symptom onset. Vaccination status was examined both as a binary variable (vaccinated or not) and for vaccines as a continuous variable (time from the last vaccine dose in weeks).

Those patients who ultimately required an invasive mechanical ventilation or expired were considered to have progressed to critically severe disease. This corresponds to levels 7, 8, 9, and 10 of the World Health Organization COVID-19 Clinical Progression Scale [21].

### 2.6. Statistical Analysis

Categorical data were summarized as frequencies (%) and numerical data as medians (interquartile range, IQR). Differences between the groups (according to vaccination) were tested using the Mann–Whitney test (numerical) or Fisher exact test (categorical variables). The association between progression to critically severe disease and a predetermined set of covariates including full vaccination status was estimated by multiple logistic regression. At the time of data collection, it was not yet known whether two patients would progress or not to critically severe disease. Therefore, two possibilities were considered: Ultimately both severe or both not severe. Results are presented as odds ratios (OR) with 95% confidence intervals (CI). To correctly include the information of time to progression to critically severe disease and 12 patients censored for ongoing hospitalization, the Fine–Gray time-to-event analysis was applied [22]. The association between vaccination and chance of discharge was also calculated by Fine–Gray analysis and supplemented by a cause-specific Cox model [22]. R statistical language (version 4.1.1) was used for the analyses, R package cmprsk was used for the Fine–Gray analysis.

## 3. Results

### 3.1. Characteristics of Hospitalized Patients with COVID-19

Among the 566 patients hospitalized with PCR-confirmed SARS-CoV-2 infection between 1 October and 23 November 2021, and assessed for enrollment, 37 (6.5%) were excluded. Thirteen were hospitalized due to medical conditions unrelated to COVID-19 and 24 were partially vaccinated against SARS-CoV-2 (Figure 1).

Among the 529 enrolled patients (median age, 65 years (IQR 53–78); 308 (58.2%) men); 175 (33.1%) were fully vaccinated. All of the patients enrolled in the study were admitted to the hospital during a period when the Delta variant was the predominant SARS-CoV-2 variant circulating in Slovenia [23]. The median time from the last vaccine dose to hospital admission was 26.6 weeks (IQR 18.7–33.1) (Appendix A). Compared with unvaccinated patients, fully vaccinated patients with breakthrough infections were older, more often immunocompromised, and had higher Charlson comorbidity index scores (Table 1).

The proportion of patients with hypoxemia at admission was comparable between vaccinated and unvaccinated groups. Most of the admitted patients (>90%) were treated with corticosteroids. Vaccinated and unvaccinated patients received specific COVID-19 treatments comparably often, except for tocilizumab, which was more often prescribed among unvaccinated patients (Table 1). At the time of evaluation on 23 December 2021, 12 patients were still hospitalized. Ten of them had already progressed to critically severe disease and the two remaining patients had been hospitalized for 31 and 40 days without progressing to critically severe disease. The median time from admission to progression to critically severe disease was 4.9 days (IQR 2–9.3) and mechanical ventilation occurred a median of 3 days from admission (IQR 1–5).

### 3.2. Association between Vaccination and Progression to Critically Severe Disease

Vaccinated patients had lower odds for progressing to critically severe disease (invasive mechanical ventilation or death) (Table 2).

The model was adjusted for age, sex, Charlson comorbidity index, and the presence of immunocompromising conditions (Table 1). Since the presence of type II diabetes, selected pulmonary and cardiovascular diseases are already part of Charlson index, we did not add them to the model in the interest of simplicity and to prevent overfitting. Apart from vaccination, lower Charlson index was also associated with lower odds for progression to critically severe disease (Table 2). Since age and Charlson index are strongly correlated variables (Spearman correlation index, 0.87), we also performed logistic regression with Charlson index excluded from the analysis. In this case, increasing age also appeared to be associated with progression to critically severe disease (OR 1.03; 95% CI 1.01–1.04; *p* < 0.001), but the model had smaller explanatory power (likelihood ratio test with *p* = 0.035). The results were almost identical when assuming that the two patients with unresolved outcome status would also have progressed to critically severe disease (Appendix A).

To include information on time to progression to critically severe disease in the analysis, Fine–Gray analysis was performed. The results concurred with those obtained with logistic regression: The incidence of progression to critically severe disease was lower for vaccinated compared with unvaccinated patients and higher for patients with higher Charlson index (Appendix A, Figure 2a).

In the Fine–Gray model, evaluating the probability of hospital discharge with a competing risk of death, fully vaccinated patients and those with lower Charlson index had a higher incidence of hospital discharge than unvaccinated patients (subdistribution adjusted hazard ratio, 1.48; 95% CI 1.19–1.83, Table 3) when adjusted for sex, age, and presence of immunocompromising conditions. The magnitude of impact on the probability of discharge is presented in Figure 2b. To better analyze the duration of hospital stay, a cause-specific Cox model with the competing event of death was fitted to the data (Appendix A). The only factor that was significantly associated with a shorter hospital stay was vaccination (HR 1.37; 95% CI 1.09–1.70).

In the subgroup exploratory analysis of vaccinated patients, we did not find a significant association between the time elapsed since receipt of the last vaccine dose and progression to critically severe disease when adjusting for age, sex, Charlson comorbidity index, and the presence of immunocompromising conditions (OR 0.99; 95% CI 0.94–1.06; *p* = 0.865).

### 3.3. SARS-CoV-2 Antibody Testing

In a subgroup of fully vaccinated patients (92/175, 52.6%), SARS-CoV-2 serologic testing was performed at admission upon the decision of the treating physicians. Antibodies to the SARS-CoV-2 spike protein receptor binding domain were detected in 90/92 (97.8%) tested patients and 61/92 (66.3%) patients had values above 2500 BAU/mL. In comparison with patients with low (≤2500 BAU/mL) antibody titers, those with high (>2500 BAU/mL) antibody titers progressed to critically severe disease less often (8/61, 13.1% vs. 10/31, 32.3%; *p* = 0.049). There was a trend towards lower Charlson comorbidity indices in those with higher titers, but this did not reach statistical significance (4, IQR 2–5 vs. 4, IQR 4–6.5; *p* < 0.058). Age was not a significant factor differentiating between the high titer (median 71 years, IQR 63–80) and low titer (median 76 years, IQR 61.5–83) groups (*p* < 0.409).

## 4. Discussion

In this study of adult patients hospitalized due to COVID-19 in a single university medical center in central Europe, fully vaccinated patients with breakthrough SARS-CoV-2 infections progressed less often to critically severe disease and had shorter duration of hospital stay than unvaccinated patients during a period with Delta variant predominance. Breakthrough cases presented a median of 26.6 weeks after the last dose of vaccine. These findings are in line with the attenuated disease severity among patients who developed COVID-19 after vaccination in the US during a period when the Alpha variant was progressively replaced by the Delta variant [6]. Vaccination against SARS-CoV-2 has previously been shown to prevent hospitalization in several studies [11,14,16,24].

In our cohort, vaccinated breakthrough cases were significantly more likely to have comorbidities compared with unvaccinated cases (Table 1). This could conceivably be due to a combination of several factors: Individuals with comorbidities might be more likely to be vaccinated [25] and/or vaccination in populations without significant comorbidities might be more protective against illnesses requiring hospitalization [14,26] and/or there might have been behavioral differences between the vaccinated and unvaccinated or between people with or without comorbidities in terms of social distancing, mask wearing, and hand washing [25].

This observational study was not designed to assess vaccine effectiveness against infection and any type of disease. Nevertheless, the preponderance of hospitalizations at longer time intervals between the last vaccine dose and admission with a median of 26.6 weeks may have been related to the decreasing vaccine effectiveness against moderate to severe disease over time, as shown previously [14,27]. Hospitalizations beyond 35 weeks from the last vaccine dose were rare, probably reflecting the fact that vaccination in Slovenia was gradually introduced in late December 2020 and the last study inclusion date was 23 November 2021. However, we cannot exclude that this drop in hospitalizations beyond 35 weeks may also have been influenced by an increasing proportion of the population receiving booster doses, and thus conceivably having increased protection from significant illness. 

Multiple factors can influence anti-SARS-CoV-2 antibody responses in vaccines, including previous disease, comorbidities, age, immune competence, and time elapsed from vaccination or disease [28,29,30]. It is still unclear to what extent antibody titers after vaccination are protective. One study found a statistically non-significant trend towards significantly poor outcomes with lower median anti spike IgG titers [26]. Although our serologic data were incomplete, results of exploratory subgroup analysis suggested that lower titers of anti-SARS-CoV-2 antibodies to spike protein receptor binding domain among vaccines were associated with a higher probability of progression to critically severe disease.

Only 12 patients eligible for the study were still hospitalized at the time of analysis. Of these, progression to critically severe disease could not be ascertained at the time of evaluation for two patients. Therefore, the great majority of eligible patients (527/529, 99.6%) were assessed for progression to critically severe disease including death up to hospital discharge. In general, the criteria for discharge vary by country and study site. Patients in our institution, including those with poor prognosis, are discharged only when clinically stable. Under these conditions, hospital discharge seems an appropriate time-point to evaluate mortality status as proposed by the WHO Working Group [21].

The majority of hospitalized patients in this study (396/529, 74.9%) had at least one comorbidity. This proportion is somewhat lower than reported in US studies (92–95%) [31,32], but higher than reported from China (25–50%) [33,34]. In addition, it probably reflects national/geographical differences in the prevalence of chronic medical conditions in the general population, but might also reflect regional differences in hospital admission criteria [21]. This study further demonstrated that a higher Charlson comorbidity index representing a higher number of underlying medical conditions increased the risk for critically severe disease. The association between individual comorbidities and higher likelihood of progression to critically severe disease has been documented in previous studies [31,32,35,36], but results were not uniform. This might be partly explained by the interplay of specific conditions or the possibility that the association between the presence of individual comorbidities and disease severity could differ by the degree of the condition, e.g., uncomplicated diabetes and essential hypertension were found to be negatively associated with the risk of death and invasive mechanical ventilation in the analysis of 540,667 patients with COVID-19 hospitalized in 800 US hospitals [32]. The results of logistic regression in our study population suggest that age *per se* was not significantly associated with progression to critically severe disease when the analysis was adjusted for underlying medical conditions, represented by the Charlson comorbidity index. Underlying comorbidities appeared to have somewhat larger impact on disease progression than age.

Highlighting the importance of breakthrough infections, the proportion of vaccinated patients in our patient cohort was higher than recorded in a large US study [13]. This may have been due to several potential factors. Increasing rates of primary vaccine uptake, waning immunity over time [37], lagging booster uptake, and potential increased infectiousness and vaccine escape may all lead to more breakthrough infections. Nevertheless, the lower odds for progression to critically severe disease (mechanical ventilation or death) in vaccinated breakthrough cases compared with unvaccinated patients in our study reinforce the value of vaccination. Furthermore, booster vaccination, although not investigated in this study, appears to decrease the incidence and severity of COVID-19 in those over 60 years of age after the emergence of the Delta variant [38].

This study has several limitations. First, due to the retrospective nature of the study, potential unmeasured confounding may have occurred, but we controlled for several relevant confounders, such as sex, age, and comorbidity status. Second, the study was limited to hospitalized patients and results could not assess the impact of vaccination on outpatient COVID-19 incidence and severity. Third, we cannot exclude the possibility of admission bias in regard to disease severity and vaccine status, but this seems less likely since the proportion of patients admitted with hypoxemia was comparable between the fully vaccinated and unvaccinated groups. Fourth, this was a single institution study limited to a particular geographic area in Central Europe with limited racial diversity and during a specific timeframe. Fifth, eligible women were slightly older than men, however the multivariate analysis corrected for this. Sixth, a relatively small sample size precluded the evaluation for the existence of differential protection against critically severe disease according to the vaccine type.

## 5. Conclusions

The results of this study suggest that vaccination remained an extremely important and effective intervention to reduce morbidity and mortality in COVID-19 during the Delta variant predominant period. When accounting for age and chronic comorbidities, previous vaccination against SARS-CoV-2 was associated with shorter hospitalizations and was a significant protective factor against progression to critically severe disease in patients hospitalized for acute COVID-19. It is unclear to what extent these conclusions might still be applicable with Omicron variant predominance and/or in the future when other SARS-CoV-2 variant(s) might emerge.

## Figures and Tables

**Figure 1 jcm-11-01191-f001:**
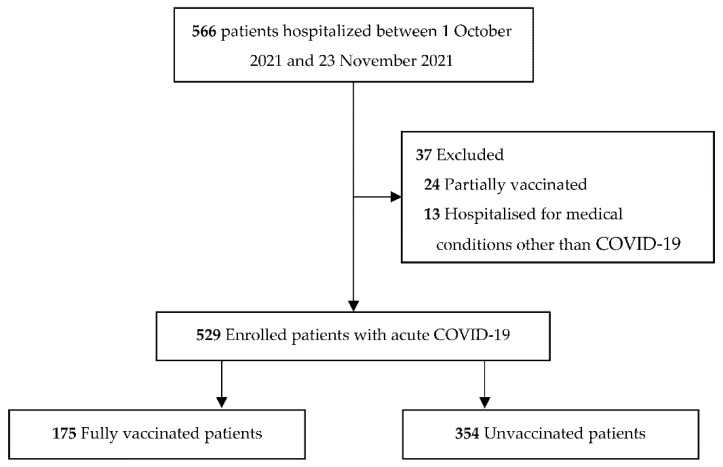
Flow diagram.

**Figure 2 jcm-11-01191-f002:**
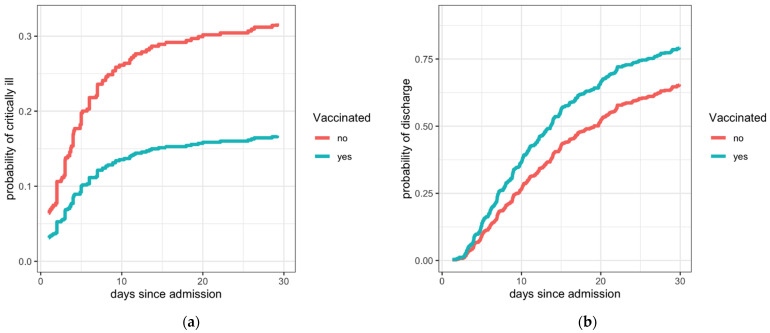
Modelled probability of progression to critically severe disease (**a**) and probability of discharge from the hospital (**b**) by vaccination status (vaccinated vs. unvaccinated) as assessed from Fine–Gray time-to-event models for two male patients with COVID-19 who were 75 years old, with Charlson comorbidity index 3, and without immunocompromising conditions.

**Table 1 jcm-11-01191-t001:** Characteristics of patients hospitalized for COVID-19 by vaccination status.

Characteristic	Fully Vaccinated	Unvaccinated	*p*-Value ^1^
	*n* = 175 (33.1%)	*n* = 354 (66.9%)	
Age	74 (64–83)	61.5 (48.2–72	**<0.001**
Men	109 (62.3)	199 (56.2)	0.191
Charlson index	4 (3–6)	2 (1–4)	**<0.001**
Chronic illnesses ^2^			
Cardiovascular disease	128 (73.1)	170 (48.0)	**<0.001**
Pulmonary disease	35 (20.0)	32 (9.0)	**<0.001**
Asthma	15 (8.6)	21 (5.9)	–
COPD	19 (10.9)	11 (3.1)	–
Diabetes type II	62 (35.4)	51 (14.4)	**<0.001**
Obesity (body mass index ≥ 30)	60 (34.3)	123 (34.7)	1
Immunocompromising condition ^3^	37 (21.1)	15 (4.2)	**<0.001**
One or more comorbidities	163 (93.1)	233 (65.8)	**<0.001**
One comorbidity	20 (11.4)	62 (17.5)	–
Resident of long-term care facility	10 (5.7)	8 (2.3)	0.071
Vaccine received			
BNT162b2	122 (69.7)	–	–
ChAdOx-1S	26 (14.9)	–	–
Ad.26.COV2.S	18 (10.3)	–	–
mRNA-1273	9 (5.1)	–	–
Therapy			
Remdesivir	21 (12.0)	34 (9.6)	0.449
Corticosteroids	162 (92.6)	331 (93.5)	0.715
Dexamethasone	140 (80.0)	258 (72.9)	0.087
Methylprednisolone	56 (32.0)	146 (41.2)	0.046
Tocilizumab	0 (0)	14 (4.0)	**0.007**
Monoclonal antibodies	5 (2.9)	9 (2.5)	0.782
Hypoxemic at admission	154 (88.0)	331 (93.5)	0.043
Critically severe disease	33 (18.9)	92 (26.0)	–
(WHO score 7–10) ^4^			
Death (WHO score 10) ^4^	24 (13.7)	46 (13.0)	–

COPD: Chronic obstructive pulmonary disease. Data are *n* (%) or median (95% interquartile range). ^1^ Due to multiple comparisons, *p*-value < 0.01 was considered significant (marked in bold). ^2^ See Appendix A for definitions of chronic illnesses. ^3^ Active cancer treatment or newly diagnosed cancer in the past 6 months (*n* = 24), active hematologic cancer (*n* = 12), previous solid organ transplant (*n* = 8), hematopoietic stem cell transplant within the last 2 years (*n* = 1), active treatment with immunosuppressive medication (*n* = 7), previous splenectomy (*n* = 1). One patient had both an hematologic and solid organ cancer. ^4^ Assessed using the World Health Organization COVID-19 Clinical Progression Scale [21].

**Table 2 jcm-11-01191-t002:** Association between prior vaccination and progression to critically severe disease (the World Health Organization COVID-19 Clinical Progression Scale 7–10) [21].

Characteristics	Odds Ratio (95% CI)	*p*-Value ^1^
Intercept	0.11 (0.03–0.37)	**<0.001**
Vaccination status (vaccinated vs. unvaccinated)	0.42 (0.25–0.70)	**<0.001**
Age	1.01 (0.99–1.03)	0.474
Sex (male vs. female)	1.47 (0.96–2.27)	0.078
Charlson comorbidity index	1.18 (1.01–1.37)	**0.035**
Immunocompromising condition present (yes vs. no)	1.04 (0.45–2.41)	0.920

CI: Confidence interval. ^1^
*p*-Value < 0.05 was considered significant (marked in bold).

**Table 3 jcm-11-01191-t003:** Subdistribution hazard ratios for discharge from the hospital.

Characteristics	Subdistribution Hazard Ratio (95% CI)	*p*-Value ^1^
Vaccination status (vaccinated vs. unvaccinated)	1.48 (1.19–1.83)	**<0.001**
Age	0.99 (0.98–1.00)	**0.037**
Sex (male vs. female)	0.84 (0.70–1.02)	0.083
Charlson comorbidity index	0.89 (0.82–0.97)	**0.005**
Immunocompromising condition present (yes vs. no)	0.87 (0.59–1.28)	0.480

CI: Confidence interval. ^1^
*p*-Value < 0.05 was considered significant (marked in bold).

## Data Availability

All anonymous data generated or analyzed during this study are available from the corresponding author upon reasonable request (dasa.stupica@kclj.si).

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
