# Peer review of "The Impact of SARS-CoV-2 Primary Vaccination in a Cohort of Patients Hospitalized for Acute COVID-19 during Delta Variant Predominance"

_jcm, 2022, doi:10.3390/jcm11051191_

Round 1

Reviewer 1 Report

The article is interesting. However, I suggest some small corrections:

  1. An Introduction is very short. The Authors should more describe vaccines against SARS-CoV-2, including their types and effectiveness. Please also cite some important articles about SARS-CoV-2 vaccines, like https://www.thno.org/v11p1690.htm and https://www.thno.org/v10p7821
  2. The Discussion has more than 2 pages and is too long. It can be shortened. 

Author Response

Response to Reviewer 1 Comments

Point 1: An Introduction is very short. The Authors should more describe vaccines against SARS-CoV-2, including their types and effectiveness. Please also cite some important articles about SARS-CoV-2 vaccines, like https://www.thno.org/v11p1690.htm and https://www.thno.org/v10p7821

Response 1: Corrections done. Please see the Introduction.

Point 2: The Discussion has more than 2 pages and is too long. It can be shortened. 

Response 2: We agree that the discussion was long and have removed two paragraphs that were admittedly somewhat tangential to the main thrust of the manuscript. Please see the Discussion.

Reviewer 2 Report

The manuscript analyzes the importance of vaccination and severity of hospitalized patients and the general outcome. The title of the article should be slightly modified since the article includes the impact of the vaccination scheme rather than the vaccination itself. Furthermore, the authors analyze the data using binary bias which is not correct. The text should be modified accordingly. Two main questions remain how many of the immunocompromised patients were fully vaccinated and how many were not? 

The subdivision of comorbidities was done correctly and the analysis is adequate. It would be interesting however to define better the comorbidities, one or more than one.

What was the anticoagulant routinely used? Why the number of patients receiving remdesivir was low? How many of the patients had a medical history of lung diseases, asthma, COPD?

Did any patient-reported side effects after vaccination?

In the discussion and conclusions, the assumptions are quite strong for a low amount of data since the statistical impact is limited, please change the text accordingly.

Author Response

Response to Reviewer 2 Comments

Point 1: The manuscript analyzes the importance of vaccination and severity of hospitalized patients and the general outcome. The title of the article should be slightly modified since the article includes the impact of the vaccination scheme rather than the vaccination itself.

Response 1: We have modified the title to specify that the study explored the impact of primary vaccination as opposed to incomplete vaccination or primary vaccination augmented by boosting.

Point 2: Furthermore, the authors analyze the data using binary bias which is not correct. The text should be modified accordingly.

Response 2: We agree with the reviewer that the WHO COVID-19 clinical progression scale, an ordinal variable, was used as a binary variable. We decided to use dichotomization because in order to define severity of disease (the primary study outcome) as objectively as possible, we used progression to mechanical ventilation or death, similar to many studies assessing association between different interventions and disease outcome/severity (JAMA. doi:10.1001/jama.2021.19499, Figure 3). Indeed, progression to mechanical ventilation or death corresponds to 7-10 in the WHO clinical progression scale, but was used as a single outcome because of its clear cut clinical relevance. Furthermore, due to the relatively small sample size and the distribution of this variable in the sample (score 4: 12 patients, score 5: 336 patients, score 6: 56 patients, score 7: 4 patients, score 8: 4 patients, score 9: 47 patients, score 10: 70 patients), our sample was too small to efficiently estimate ordinal logistic regression with seven categories.

Point 3: Two main questions remain how many of the immunocompromised patients were fully vaccinated and how many were not? 

Response 3: Out of 52 immunocompromised patients, 37 (71.2%) were fully vaccinated and 15 (28.8%) were unvaccinated. Please see Table 1.

Point 4: The subdivision of comorbidities was done correctly and the analysis is adequate. It would be interesting however to define better the comorbidities, one or more than one.

Response 4: In order to quantitate patients' comorbidities, we used Charlson comorbidity index and added information in Table 1 to better define the comorbidities as proposed by the reviewer. Please see Table 1 and Supplementary Table 1.

Point 5: What was the anticoagulant routinely used?

Response 5: In our institution, dalteparin is routinely used for thromboprophylaxis in patients with COVID-19. Patients received thromboprophylaxis, so this parameter did not introduce bias to the intergroup analysis.

Point 6: Why the number of patients receiving remdesivir was low?

Response 6: This was a retrospective observational study. Therefore, we can only speculate that the low rate of remdesivir use was due to historical lack of evidence that remdesivir importantly impacts treatment outcome, unless given early in the course of infection. This led to a conditional recommendation against its use in hospitalized patients issued by the WHO in late 2020. Several recent studies suggest that remdesivir may be more beneficial than previously thought in different clinical scenarios and its place in therapeutics is evolving. Since only adults hospitalized due to COVID-19 were assessed in our study, those outpatients not requiring supplemental oxygen, diagnosed within 7 days after symptom onset and given remdesivir because of increased risk of developing severe COVID-19 were not included in our study. Furthermore, there was no statistically significant difference in the use of remdesivir between the vaccinated and unvaccinated groups; therefore, no remdesivir-related bias was introduced into the analysis.

Point 7: How many of the patients had a medical history of lung diseases, asthma, COPD?

Response 7: This information was added. Please see Table 1.

Point 8: Did any patient-reported side effects after vaccination?

Response 8: Unfortunately, information on side effects after vaccination occurring a median of 26.6 weeks before hospital admission was not available in medical charts. Side effects to vaccination were beyond the scope of our investigation.

Point 9: In the discussion and conclusions, the assumptions are quite strong for a low amount of data since the statistical impact is limited, please change the text accordingly.

Response 9: Corrections done. Please see the Conclusions.

Round 2

Reviewer 2 Report

The authors have responded to all the queries. The manuscript is suitable for publication